# Effects of a Hybrid Program of Active Breaks and Responsibility on the Behaviour of Primary Students: A Mixed Methods Study

**DOI:** 10.3390/bs12050153

**Published:** 2022-05-18

**Authors:** José Francisco Jiménez-Parra, David Manzano-Sánchez, Oleguer Camerino, Queralt Prat, Alfonso Valero-Valenzuela

**Affiliations:** 1Faculty of Sport Sciences, University of Murcia, 30720 Murcia, Spain; josefrancisco.jimenezp@um.es (J.F.J.-P.); avalero@um.es (A.V.-V.); 2National Institute of Physical Education of Catalonia (INEFC), University of Lleida, 25192 Lleida, Spain; ocamerino@inefc.udl.cat (O.C.); qprat@inefc.es (Q.P.)

**Keywords:** physical activity, innovative methodologies, values education, teacher satisfaction, teaching strategies

## Abstract

Schools are ideal environments to promote healthy lifestyles and teach values among students. In this sense, the present study aims to verify the result of an Active Break program (AB) within the Teaching Personal and Social Responsibility (TPSR) Model in the school environment. The sample consisted of two teachers/tutors from the sixth year of Primary Education and 51 pupils, aged between 11 and 13 years, who were divided into an experimental group (*n* = 26) and a control group (*n* = 25). The intervention program lasted 3 months, in which the hybridised methodology was applied during 100% of the weekly classes, computing a total of 156 sessions by the end of the study. It was a quasi-experimental study design that used a mixed methodology combining a systematic observational analysis with semi-structured interviews. The results showed an evolution in the behaviour of the teacher from the experimental group from a controlling style to one centred on the transfer of autonomy, while the teacher from the control group primarily used strategies based on the imposition of tasks and the establishment of organisation, which caused an increase in disruptive behaviours among students. We conclude that the program is adaptable to Primary Education and can be extended to any educational environment to improve the classroom climate and attract the attention of students and, finally, allows for the promotion of new teaching strategies.

## 1. Introduction

Pre-adolescence and adolescence are considered the periods of life in which externalising problems (e.g., antisocial behaviour, aggression, bullying or violence) and internalising problems (e.g., shyness or social anxiety) related to prosocial behaviours begin to increase significantly [1]. These behaviours can be understood as voluntary behaviours aimed at benefitting others and play a key role in creating positive interpersonal relationships and maintaining personal and social well-being [2]. Therefore, greater attention should be paid to the educational values that young people receive from the family and educational environment.

These types of behaviours are aggravated by the increase in sedentary behaviours of children and adolescents, putting the health of young people and the maintenance of health in the medium and long term at risk [3]. In this regard, physical inactivity has been considered a key element in the development of childhood obesity [4]. The World Health Organization (WHO) [5] has established recommendations aimed at children and adolescents between 5 and 17 years of age, indicating that they should accumulate a minimum of 60 min per day of moderate and/or vigorous physical activity (MVPA), or 3 days per week of vigorous activity with high musculoskeletal involvement, in addition to reducing sedentary time, especially that dedicated to the use of screens. Recent studies have shown the benefits of implementing programmes that promote physical activity (PA) during school hours as a preventive measure against sedentary children and as a formula to combat obesity and possible associated metabolic pathologies [6,7].

In short, recent studies suggest the promotion of research based on interventions in schools that seek the development of values to mitigate school violence by influencing the reduction in social conflict [8], as well as the implementation of research that promotes PA during school hours as a measure to prevent children’s sedentary lifestyles in order to combat obesity and possible associated metabolic pathologies [5].

### 1.1. Active Breaks at School

In this context, schools become ideal environments to promote PA practice because children and adolescents spend a large part of their time at school and because they are accessible to the public regardless of age, gender, family socioeconomic status or ethnicity [9]. Moreover, in this environment, there are multiple opportunities for students to be physically active during the school day, such as physical education (PE) classes, active transportation, recess and active breaks [10,11].

Evidence shows that programmes where active periods take place in the school setting can be effective in increasing students’ fitness levels [12] and reach up to 50% of the daily PA recommendations set by the WHO [13]. However, it should be noted that schools have difficulties in implementing these PA periods due to two factors: the predominance of learning areas considered key [6] and the resistance of educators to integrate PA into the classroom, presumably due to greater disorganisation [14]. However, scientific evidence suggests that continuous PA has a motivating and positive effect on students’ classroom behaviour [15,16].

PA in the classroom is a promising strategy for students around the world to be active during school hours [16]. Given this fact, the so-called active breaks (AB) appear, which consist of the incorporation of PA by teachers in the classroom timetable through the inclusion of integrated physical exercise or the implementation of PA for short periods of time, working with curricular content (active breaks based on the curriculum) or without the presence of curricular content [17]. Given this evidence, ABs show a positive impact on students’ behaviour in class [17,18] on cognitive and academic performance [18,19,20] and a satisfactory perception of teachers and students [21].

### 1.2. Teaching Personal and Social Responsibility Model (TPSR)

In addition to the concepts mentioned above, programmes aimed at promoting personal and social responsibility have a positive impact on the climate of coexistence in schools, taking advantage of the interactivity and the emotional and affective nature that teaching in values allows [22]. In order to educate in values, one of the fundamental aspects to consider is that of the pedagogical strategies used by teachers so that students acquire this education [8]. In this sense, TPSR [23,24] which seeks to promote responsibility and autonomy, has shown very positive results in promoting values in schoolchildren [25] and in improving the quality of teaching [8].

The beginnings of this pedagogical model were aimed at working values through PA and sport with socially disadvantaged young people [24]; however, over the years it has expanded its range of application to other sporting contexts such as extracurricular activities or training and competitions to develop different psychosocial aspects such as respect, self-esteem, empathy, effort, autonomy and cooperation [8,26]. Recent studies have suggested the promotion of research-based interventions in schools that aim to develop values and mitigate school violence [27], as well as the promotion of PA during school hours as a preventive measure against sedentary lifestyles [28]. In the same way, its implementation has been carried out not only in the sporting field, specifically in PE [8], but in all subjects of the Primary and Secondary Education curriculum [29,30]. However, there is still no study which has hybridised both methodologies, looking for its viability and the impact in student values and PA.

Therefore, the main objective of this study was to verify the evolution in the behaviour of teachers and students during the application of a hybrid educational programme based on the use of AB within the TPSR intervention, checking how active methodologies, such as the one reflected in this study, can improve student behaviour. A second objective was to identify teachers’ perceptions regarding the implementation of this innovative and hybrid methodology and its impact on the promotion of values and the increase in students’ PA.

## 2. Materials and Methods

### 2.1. Design

In this quasi-experimental study, a mixed methods approach [31] was applied, complementing the systematic observation methodology to analyse teachers’ strategies and communication during the implementation of a programme, with semi-structured interviews to identify teachers’ perceptions regarding the results of the sessions.

A nomothetic, follow-up and multidimensional observational design was applied [28] observing the behaviour of all students and the individual behaviour of teachers, considering their evolution throughout the administration of the intervention programme in a temporal process, and analysing various relevant dimensions in the behaviour of students and teachers, reflected in a multiplicity of dimensions in the observation instrument.

### 2.2. Participants

The sample was purposive and by convenience, selection for registration included the following:

Students: the sample consisted of a total of 51 students distributed in two groups in the sixth year of the last stage of Primary Education, aged between 11 and 13 years (M = 11.73; SD = 1.73), with a medium–low socio-economic level. None of them had previous experience with AB and TPSR and were divided into two groups, the control group (CG) and the experimental group (EG), with 25 and 26 students, respectively.

Teachers: two teacher/tutors with similar teaching experience (between 10 and 12 years), each taking responsibility for one group: (a) the CG teacher who maintained the methodological strategies based on conventional teaching, and (b) the EG teacher who developed the hybrid programme based on TPSR and AB; the latter received training in the innovative teaching strategies of both methodologies. Inclusion and exclusion criteria for the study were (a) that the students had not previously been taught in either of these methodologies and (b) that they did not have any physical or mental disabilities that would prevent them from participating in the study.

### 2.3. Instruments

#### 2.3.1. Instruments to Verify the Implementation of the Programme

Registration to evaluate the use of the TPSR (TARE 2.0) [32] involved the use of the Spanish version validated by Escartí et al. [32]. The first section was used which was dedicated to assessing responsibility-based teaching strategies. It consists of an observation method that uses interval data collection to record the strategies applied by teachers implementing the TPSR. Two observers analysed the presence or absence of the described categories during 3-min periods of continuous monitoring. The total interobserver agreement was 85.2%. Similarly, intra-observer reliability was calculated with the analysis of the same video by the observers after a certain period of time (7 days), before the start of the research, with an agreement of more than 90% for each of them separately. It consisted of a system of 9 categories: (1) modelling respect, (2) giving expectations, (3) giving opportunities for success, (4) encouraging social interaction, (5) assigning tasks, (6) leadership, (7) giving choice and voice, (8) role in evaluation and (9) transfer. An analysis of the EG sessions was carried out every 15 days, providing them with a feedback report on the aspects to improve and highlight in terms of the use of TPSR.

The instrument to assess the active breaks (IEDA) [33] was used to evaluate teaching strategies based on AB. The IEDA was created and reviewed by three experts in AB interventions to seek consensus on the different categories after it was used in different recorded sessions. It was composed of 8 categories: (1) class disruption, (2) movement, (3) academic content, (4) social interaction, (5) structure, (6) motivation, (7) participation/activation and (8) back to class. It was verified by two observers using dichotomous responses (yes/no), depending on whether or not the strategies were applied during the PA integration period in the classroom (5–10 min). Inter-observer reliability was 89.1% and intra-observer reliability was over 95%.

#### 2.3.2. Instruments for the Analysis of Teacher Performance

The observation system for teaching-oriented responsibility (OSTOR) [34], “SORPS” [35], was used which was validated by expert observers [36]. It consists of six dimensions or criteria and several categories in each of them (Table 1).

#### 2.3.3. Evaluation Instruments and Teachers’ Perceptions

The semi-structured individual interview involved a qualitative analysis being carried out through a semi-structured interview [37] at the end of the intervention process, with the aim of finding out the teachers’ perceptions and opinions about the experience of implementing the hybrid programme in the different educational areas. For this purpose, 11 categories were created, which in turn had different questions: methodological resources (“Do you feel you have more tools to educate schools or children about behaviours that disrupt coexistence?”), teaching experience (e.g., “Do you consider that you are sufficiently trained to implement the TPSR? And the AB?”), methodological difficulties (e.g., “What are the main problems you have encountered?” “What are the main problems that have arisen?”), satisfaction (“How do you feel about implementing TPSR? And AB?”), innovative aspects (“What are the most innovative aspects that you think TPSR is bringing to your classes? And AB?”), curricular adaptation (e.g., “Do you think that some content can be more effective?” “Do you think some content might be more appropriate for the application of TPSR? What about AB?”), advantages of the methodology (e.g., “Do you think that TPSR really works for the incorporation of values, attitudes and socially appropriate values?”), student opinions (“Do you think that the students’ opinion is positive regarding the hybrid methodology?”), suggestions for improvement (“What do you think can be improved in the application of this hybrid methodology?”), future implementation (“Would you apply or reapply this hybrid methodology in your classes in future years?”) and other aspects to consider (“What would you like to comment on that has not been said so far?”).

### 2.4. Procedure

After the management team and two teachers agreed to carry out the programme in a school in the Spanish region of Murcia, informed consent was obtained from the parents and/or guardians of all participants (confidential data, participation in the study and filming of sessions), in accordance with the ethical guidelines on confidentiality and anonymity of the Research Ethics Committee of the University of Murcia (3207/2021).

#### Intervention

The study lasted for three months (12 weeks) in which a programme was applied during 100% of the classes (12 weekly sessions), for a total of 156 sessions throughout the intervention. The two participating teachers followed the curriculum of the Spanish education law and the syllabus of the school, but each one carried it out with a different methodology in the respective areas of Spanish Language, Mathematics, Social Sciences and Natural Sciences (Table 2). A SONY HDR video camera was used for the recordings; decreasing the reactance bias by recording sessions prior to the start of the programme, in order to favour the appearance of spontaneous behaviours [28]. All recordings were considered optimal: nine EG teacher sessions and eight CG teacher sessions with a duration of 55 min.

In order to provide greater clarity and representativeness of the intervention, Table 2 shows the contents worked on in each of the areas during the implementation of the programme and the methodological strategies used depending on the group.

For the *control group (CG)*, the CG teacher used a teaching style based on direct command, a strategy typical of the traditional school [38]. In this context, the teacher is positioned as the key element in the teaching–learning process whose main function is based on being a transmitter of content, with a large volume of information, in an imposing and authoritarian manner [39], while the student is in a passive role and a function limited to reproducing the established tasks.

For the *experimental group (EG)*, the EG teacher implemented an innovative programme based on the hybridisation of TPSR and AB through the use of innovative classroom strategies. The hybrid EG sessions followed the format proposed by Hellison of assigning tasks, giving opportunities for success, defining roles, providing positive and continuous feedback and incorporating the 10 min of AB in their structure: (1) awareness: the teacher interacted with the students to create bonds with them, set expectations, explained the academic objectives and the value of the session in terms of the level of accountability, (2) responsibility in action: inclusion of responsibility strategies in the different tasks and where the ABs were put into practice with PA for a set time (5–10 min), (3) group assembly/meeting: the teacher asked a series of open questions to the students to check that the objectives of the session and the knowledge and contents had been assimilated and (4) self and co-evaluation: the students assessed whether they had met the objectives set by the teacher at the beginning of the session, as well as the performance of the teacher and their peers using the thumb technique.

In accordance with the development of the TPSR, the intervention followed a progression in which the teacher incorporated a higher level of responsibility every three weeks, with all students reaching the maximum levels of the TPSR. The levels working with the model, in a progressive and cumulative way, were (1) respect for oneself and for the rights of others, (2) participation and effort, (3) personal autonomy, and (4) leadership and helping others. Level 5, called “transfer”, was developed simultaneously with each of the levels in order to transfer the values to everyday life situations by introducing examples of transfer at the end of the sessions according to the contents developed (Table 2 and Table 3). Table 3 describes the activities and practical task examples that were carried out in the EG intervention in each one of the curricular contents.

More specifically, ABs were developed after 20–25 min of class when teachers observed a decrease in attention on the part of the students or when students requested it by mutual agreement with the teacher. The following methods were used to perform these AB: (1) Tabata routines [40]: high intensity training combining 5 to 7 exercises (table push-ups, lunges, squats, squat jumps, squat jumps, multiple jumps, sit-ups, chair raises, adapted burpees, etc.) of 15 to 20 s, with rest periods of 10 s [41,42]; (2) physically engaging active videos or Brain Breaks Videos [43]: audiovisual resources on digital whiteboards that generated a motor response in students; and (3) cognitively engaging AB [44]: activities designed to work on and reinforce curricular content (Table 2 and Table 3).

### 2.5. Statistical Analysis

Two observers were initially trained as experts in innovative methodologies and observation techniques, who achieved an inter-observer reliability agreement in the recording of the OSTOR observation instrument of 86.7% and 95%, respectively. T-pattern detection and analysis (TPA) is a technique implemented in Theme v.6 software [45] in order to detect recurring synchronic and sequential patterns of behavioural events. Temporal patterns (T-patterns) represent sequences of events characterised by statistically significant constraints among the interval length separating them with the filters of (a) frequency of occurrence equal to or greater than 3; (b) significance level less than 0.005, percentage acceptance of a critical interval due to 0.5% probability; and (c) redundancy reduction setting of 90%.

The interview was analysed with ATLAS-Ti software. V.7. 1.3 [46], recorded on video and treated in multimedia format and stored in a single file (Hermeneutic Unit). We coded each of the sentences according to the different aspects of the main analysis, reflected different considerations about this intervention. Specifically, we reflected the answers to each of the questions: (a) adequacy of the content, (b) innovative aspects of the methodology, (c) teaching difficulties in its implementation, (d) building values with the model, (e) adequate training, offers autonomy to the teacher, (f) future implementation of the methodology, (g) importance of teaching experience, (h) importance of continuous training, (i) need to adapt to students’ interests, (j) positive opinion of students on the methodology, (k) proposals for improvement.

## 3. Results

The results for the behaviour patterns (T-patterns) of the two teachers are presented first, followed by the results for the interview with the EG teacher. These results are shown by means of the technique of T-pattern detection and analysis (TPA) described above. As it is shown in Figure 1, T-patterns were obtained by mean Theme v.6. software showing figures of dendograms of the behaviours that conform to a recurrent T-pattern. This allows representative sequences of events characterised by statistically significant constraints among the interval length separating them and can be read sequentially from top to bottom.

### 3.1. Pedagogical Performance of Teachers

The two teachers showed opposite behaviours, represented by the T-patterns of Figure 1, and it should be read as:The EG teacher suggested (SUG) the intervention of the students, giving them autonomy (AUT) and also giving them a positive evaluation (POS). The teacher’s combination of positive evaluations (POS) and suggestions (SUG) led to the autonomous response (AUT) of the students. The teacher’s proposal of new opportunities for success (PRO) also allowed for attention and effectiveness in achieving the session’s objectives.The CG teacher imposed task instruction (IMP) and set them by himself (EST), provoking a reproductive response from the students (REP) without giving alternatives. His controlling style with negative evaluations (NEG), the establishment of the organisation (EST) and the imposition of instructions (IMP), provoked an unbalance and a misaligned (UNB) and reproductive response (REP) from the students, which forced the teacher to redirect the classes (RED) through negative evaluations (NEG).

Figure 2 shows the Event Time Plot of all the events registered throughout the intervention. The x axis is the temporal line of frames separated by periods of 5 min (almost 8.000 video frames). The y axis refers to the codes of behaviours observed (e.g., number 41 concerns SUG and AUT (suggested and autonomy)). The solid vertical blue lines separate the sessions and the vertical solid line in the centre separates EG and CG. We highlighted rectangles to emphasise the codes of behaviours that appear, whether it be more density or a lack of density, in order to offer a more detailed analysis of the performance of the two teachers.

The representation of all sessions is shown and in the middle area of Figure 2, a higher concentration of behaviours of the EG teacher conformed by suggestion (SUG) and autonomy (AUT) of the student can be seen that were more numerous than those of the CG teacher on the right side. In line with this, we can verify the accumulation of behaviours in all sessions of the CG teacher, represented on the right side and in the lower area, of imposing instructions (IMP) and establishing organisation (EST) in the activities that generated reproductive responses (REP) from the students. In the upper area, the positive assessment (POS) of the EG teacher and the negative evaluation (NEG) of the CG teacher and the maladjusted behaviours (DES) of the students in this group are represented. 

To conclude the T-Pattern analysis, we compared the results of the implementation of strategies associated with each methodology during the development of the programme, showing different profiles of chained and typical behaviours (Figure 3 and Figure 4) between both teachers, as seen in the previous dendogram (Figure 1). On the one hand, Figure 3 reflects the behaviour patterns associated with more innovative and active teaching strategies in which the teacher guides the students through the transfer of autonomy.

In the first place (upper dendogram), we observe a combination in which the teacher suggests (SUG) the intervention of the students in class reinforcing autonomous behaviour (AUT) in the students, which is preceded by a positive evaluation (POS). Secondly (bottom dendogram), it can be seen that, after carrying out the previous behaviour pattern, it is again preceded by an innovative behaviour on the part of the teacher in which he/she suggests (SUG) the students’ participation by reinforcing their autonomy (AUT) through the formulation of proposals (PRO) and new opportunities for success and options.

From these results, it can be highlighted that the relationship of these patterns occurs only in the teacher of the EG. Moreover, it is observed that the relationship begins to be significant from the middle of the intervention and that it is accentuated in the last weeks of the study.

On the other hand, Figure 4 reflects the behaviour patterns associated with more directive teaching strategies where students simply reproduce the teacher’s instructions.

In the dendogram at the top, a combination is observed in which after mismatched behaviour (UNB) of the learners, the teacher makes a negative evaluation (VAN) of their behaviour, which is preceded by the imposition (IMP) of the task, the establishment (EST) of the organisation and the reproductive response (REP) of the learners. The dendogram at the bottom shows that after the imposition (IMP) of the task and the establishment (EST) of the organisation of the task, it is preceded by another pattern of imposition (IMP) and establishment behaviour in which the students simply reproduce (REP) what is established by the teacher.

In addition, there are other very interesting results that can be extracted from Figure 4. The significant relationship of these behaviour patterns occurs more frequently in the CG and remains stable throughout the intervention. However, it is observed that the EG teacher applies these combinations at the beginning of the study, but then they are gradually reduced until they disappear at the end of the research. This event is associated with and can be compared to the dendograms in Figure 3. Initially (first weeks), the teacher starts with a more direct teaching style, although from the middle of the intervention onwards, he modifies it for more innovative behaviours based on the transfer of responsibility and autonomy to the students, being more significant at the end of the intervention.

### 3.2. Teacher Evaluation and Perception

The information extracted from the interviews was grouped into four families. The information is presented according to the categories of analysis, showing in each one, data (extracts) related to the intervention group. The categories were as follows:

Strengths and weaknesses of the methodology. This category had 52 extracts and relates to all those aspects of the methodology that represent an advantage or a disadvantage from the teacher’s point of view. The extracts that make up the strengths had 44 mentions and the code for weaknesses had 8 mentions.

For instance, we found the mention, “A mother’s discomfort with the formation of an TPSR strategy to resolve conflicts in the classroom. Because she said her child was autistic and it didn’t suit her”.

On the other hand, the second category was named “importance of teacher training and experience” with 17 extracts, which was related to “adequate training, provides teacher autonomy” and “importance of teacher experience”. Most parts of the extracts described aspects about the importance of in-service training such as “Honestly, I consider as key aspects the calls and meetings we have had for in-service training, because maybe you understand it somehow during the initial training and then it was not done exactly like that”.

The third category was called “methodological innovation” with extracts for didactic/teaching resources and innovative aspects of methodology. Some statements made by the teacher on this topic can be highlighted, such as “It has allowed me to divide or structure the session a little more. And then, the introduction of the AB in its entirety has been something totally new for me because I had heard about them, but I didn’t know how to apply them or anything. So it has been a novelty in my classes”. Didactic resources/teachers was mentioned several times where the teacher reflected on all the resources that the methodology provided to improve their teaching, such as “Seeking feedback with the students was something that I found very interesting, it is an aspect to take into account and that I will possibly implement in the future”.

Finally, the fourth category was called “Adaptation to the curriculum and the students”, including aspects about “adaptation of the contents” and “need to adapt to the interests of the students”. One of the comments made by the teacher was “I think that it can be adapted to all content as it is one of the principles of TPSR, that it has transfer to the street and that it is transversal and interdisciplinary between different subjects, and as for the AB, I think that it can be adapted to any content and educational area”.

## 4. Discussion

The main objective of this study was to analyse the behaviour of teachers and students during the implementation of a hybrid educational programme based on the use of AB within TPSR. A second objective was to identify the teacher’s perception of the implementation of this hybrid methodology and its impact on the promotion of values and the increase of students’ PA.

Regarding the first objective, the OSTOR results show that the EG teacher mainly adopted a behaviour pattern oriented towards reinforcing students’ responsibility and autonomy, both aspects being two of the fundamental pillars of TPSR [24]. On the other hand, in the CG, it was observed that patterns related to the imposition of the task and the establishment of organisation predominated, with greater reproduction and less capacity for autonomy in decision making, being more related to conventional styles [47]; these differences are consistent with different studies such as that of Gordon [48] and Prat et al. [35] where after applying the TPSR in PE, behaviours related to autonomy and responsibility were reinforced, while the teacher who used a traditional methodology applied the imposition of the task and organisation, provoking a reproductive response from the students. Likewise, the study by Valero-Valenzuela et al. [49] that hybridised TPSR with gamification found results very similar to those of the present study, where the teacher promoted the transfer of responsibility and autonomy to students during classes.

The behaviours of the EG teacher promoted the transfer of responsibility and autonomy inherent to the TPSR strategies, showing that the teacher’s performance is a determinant on the behaviour of the students, as already advanced by López et al. [15] in their study with Primary school students. These results were similar to those of Camerino et al. [34] and Valero-Valenzuela et al. [50] where through continuous monitoring and training, applying the TPSR achieved a reduction in maladaptive and disruptive behaviours, not only in PE, but also in other subjects and educational areas. In addition, new adaptive patterns related to the establishment of session and task goals were generated, as is also the case in the study by Prat et al. [35]. At the same time, we highlight that the teacher’s behaviour improved throughout the intervention, probably due to the continuous training carried out, necessary for its evolution, through weekly meetings with the research group. This supports the importance of continuous training in this type of intervention [51,52]. This type of training is necessary for patterns to evolve, not only towards more adaptive behaviours, but also towards a reduction in maladaptive behaviours [34,52].

As a second objective, the teachers’ positive perceptions of the hybrid programme were assessed, highlighting the “innovative aspects of the methodology”, including the proximity to the students and the organisation of the sessions. The use of TPSR makes it possible to structure the sessions, facilitating the work of teachers and students [27], helping to counteract the difficulties of physically active classes [53,54]. Similarly, the adaptability, interdisciplinarity and complementarity that may exist between curriculum areas to apply these types of programmes is based on TPSR and AB, both in Primary Education [30,55,56] and in Secondary Education [29,57,58]. Furthermore, studies such as Dinkel et al. [53], Manzano-Sánchez et al. [59] and Martin and Murtagh [54] state that teachers consider the activities to be easy to implement, adaptable and feasible, but these results do not coincide with the studies by Martos et al. [60] on TPSR and Watson et al. [10] on AB, which reflect teachers’ difficulty in applying innovative methodologies due to lack of time.

The EG teacher stated that this new way of teaching has allowed him to feel more satisfied on a personal level, as well as to perceive greater satisfaction in the students. In this sense, it can be affirmed that it follows the line of other research such as that of Sánchez-Alcaraz et al. [8] highlighting the positive perception of the increase in teaching resources by applying TPSR; and the studies by Martin and Murtagh [54] and Riley et al. [56] regarding teacher satisfaction by applying an AB programme, due to the didactic resources provided by this methodology [22]. Other studies [54,61] agree that physically active classes provide an opportunity to promote creativity, reflection and teacher training, offering new resources that increase the enjoyment of their work [56]. In addition, the teacher felt noticed how the students had acquired and learned new attitudes and values essential for the improvement of motor development, PA level, concentration, attention and learning in an autonomous and self-directed way [61], as well as improving the classroom climate as indicated by Sánchez-Alcaraz et al. [8].

As negative aspects were highlighted in the interviews, the EG teacher perceived difficulties especially at the beginning of the intervention due to the uncertainty of not knowing how to act, although he stated that these errors were reduced with the familiarisation and feedback provided by the research group during the continuous training. Another weakness found is related to the difficulty in performing group tasks during AB, as also pointed out by Ayuso [22] and Watson et al. [10] in their studies. This was due to problems of space and organisation. In addition, the teacher indicated proposed possible improvements to this programme based on the need for its implementation by a greater number of teachers in schools, as well as proposing more longitudinal studies. This proposal was also made by teachers in other studies [56] with the aim of generating greater positive effects on students. Finally, the incorporation of more intense and shorter physical exercises is also considered a factor to be improved. In this sense, Watson et al. [10] reflect on these same results in the statements of the teachers who participated in their study.

Although the combination of both methodologies shows positive effects on teachers’ and students’ behaviour patterns and teacher perceptions, the results and conclusions of this study should be considered and interpreted with great caution because the research has limitations of relevance. First, the study sample is too small to generalise the results and the methodology has only been applied in Primary Education and in specific curricular areas such as Mathematics, the Spanish language and Social and Natural Sciences. Future studies should look at larger samples involving as many teachers as possible, and cover other educational stages as well as other subjects in order to obtain more conclusive results on the effectiveness of the programme. Second, the duration of the intervention was three months, which is below the suggestions of expert authors in TPSR [8,57,62] and AB [18], who have indicated the need to implement longitudinal programmes, longer than four months, to have positive effects on participants. Third, it should be noted that it would have been interesting to consider quantitative results through the analysis of questionnaires and accelerometers, as carried out by Valero-Valenzuela et al. [49] and Watson et al. [10], respectively, to complement the results obtained.

Taking into account the aspects mentioned above, changes in the implementation of innovative strategies related to TPSR and AB could remain a promising process for methodological change in education. This, together with the teacher specific professional development and continuous training based on directive teaching, could help teachers to renew their teaching methods. In this study, it has been found that the predominant pedagogical behaviours with this innovative methodology (based on the incorporation of AB into the TPSR structure) are those that suggest student intervention through positive teacher appraisal and transfer of responsibility, which generates autonomous student responses and a reduction in maladaptive behaviours. In contrast, a conventional methodology generates less participatory and more disruptive reproductive responses from students.

To conclude, we highlight the relevance of using observational methodology with the technique to detect T-pattern analysis (TPA) [37] that allows the refinement of substantial results, as we have obtained in this study. As the TPA has been proven to be very useful in related subjects such as teachers’ verbal and non-verbal communication [63,64] and the adhesion to physical activity [65,66,67], we have improved this study by spreading the objectives for study related to those areas.

It is concluded that this programme, to the best of our knowledge, is the first study that contemplates the hybridisation of these two methodologies, standing out for its innovative character to promote autonomy and active participation in the classroom, as well as for its applicability and adaptability to any context. This fact could allow for the extension of the methodology to any educational area to improve the classroom climate and capture the attention of students. Finally, it has generated greater satisfaction among the teaching staff as they have observed progression and received positive feedback from the students.

## Figures and Tables

**Figure 1 behavsci-12-00153-f001:**
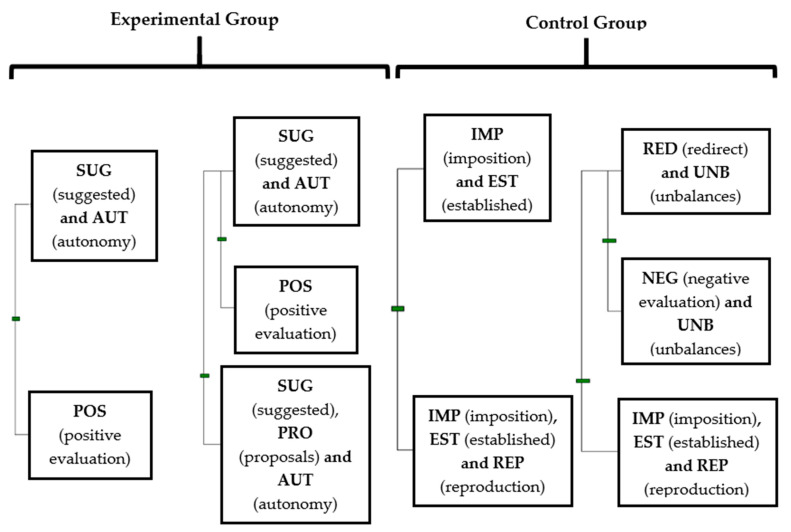
Representative dendrogram of teachers’ performance, experimental (EG) and control (CG), and students’ response in the analysed sessions of the intervention.

**Figure 2 behavsci-12-00153-f002:**
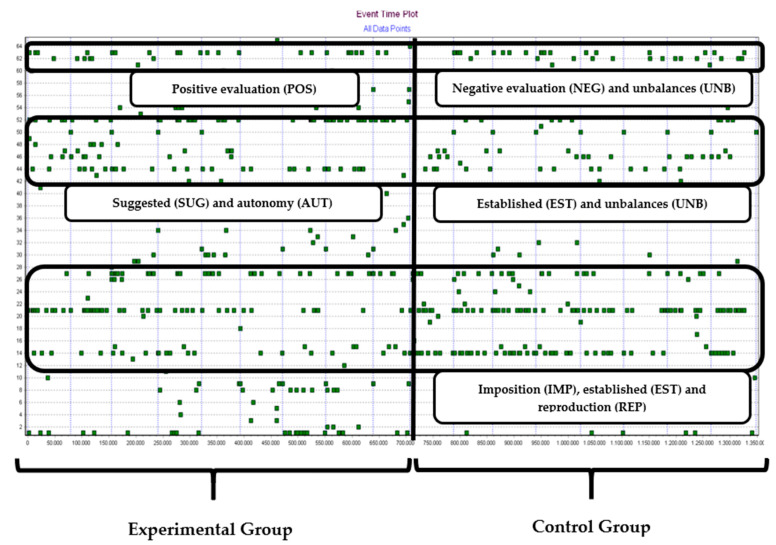
Event Time Plot of EG and CG behaviours throughout the intervention.

**Figure 3 behavsci-12-00153-f003:**
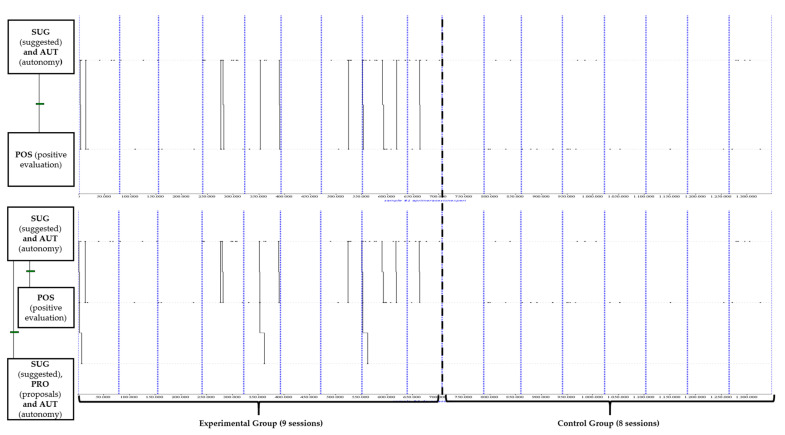
Dendogram representing the comparison of temporal patterns of teachers (EG and CG) associated with innovative teaching behaviours (guide to learning), and student responses.

**Figure 4 behavsci-12-00153-f004:**
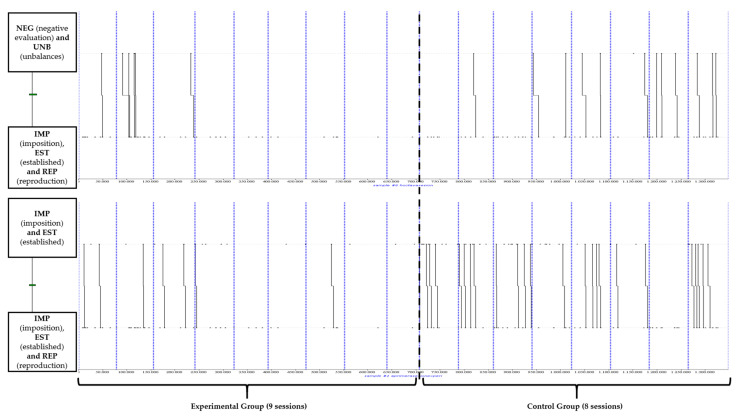
Dendogram representing the comparison of temporal patterns of teachers (EG and CG) associated with traditional teaching behaviours (direct command), and student responses.

**Table 1 behavsci-12-00153-t001:** Observation system for teaching-oriented responsibility “OSTOR”.

Criteria	Category	Code	Description
Expectations	Objective of session	OBS	Prospects and aims of the session
	Objective of task	OBT	Prospects and aims of the task
Explanations	Imposition instructions	IMP	Without the possibility to include changes
	Shared	SHA	Allows the proposals to be decided collectively
Organisation	Established	EST	Establish spaces and materials directively
	Distribution of function	DIS	Functions and roles are allocated
	Suggested	SUG	Teachers pose opportunities for students’ interventions
	Combination of organisations	COO	Combining several organisations at the same time
Task Adjustaments	Negative evaluation	NEG	Criticising and negatively reprimanding students
	Redirect	RED	Correct students’ responses
	Positive evaluation	POS	Encourage and motivate the students
	Proposals	PRO	Formulate new options to be successful
	Self-assessment questions	SAQ	Ask questions for reflection
	Combination of modulations	COM	Combines several modulations
Students’ Responses	Reproduction	REP	Replicate tasks or situations
	Unbalance	UNB	Disarranged or disordered responses
	Autonomy and leadership	AUT	Drives initiatives in an autonomous manner
	Self-assessment	SAS	The student evaluates its own performance
Session Summary	Guided summary	GUS	The teacher summarises the session
	Shared summary	SHU	The students take part in the sessions summary
	Nonexistent summary	NSU	The sessions end without being summarised

**Table 2 behavsci-12-00153-t002:** Weekly content example and methodological strategies used by each group.

	Methodology and Teaching Strategy						
	EG		CG	Educational Areas and Contents (Both Groups)			
Weeks	Strategies and Level of TPSR	Methods of AB	Conventional Teaching	Mathematics	Spanish Language	Social Sciences	Natural Sciences
1–3	Introduction to TPSR, L1 and L5. Dispute resolution and liability contract	Tabata and curricular AB routines	Direct command. Imposition of individual tasks.	Powers and roots	“Join the party”: determiners, compound words, etc.	The relief	Living beings
4–6	L2 and L5 (L1 reinforcement). Problem solving tasks through effort	Tabata routines and active videos	Direct command. Imposition of individual tasks.	Fractions	“Observe and act”: verb, tilde and types of novels	The water	Energy and matter
7–9	L3 and L5 (L2 reinforcement). Activities to reinforce independence and decision making.	Tabata routines, active videos and curricular ABs	Homework assignments. Establishment of activities, passive learner.	Decimal numbers	“Geniuses”: creativity	European citizenship	Machines
10–12	L4 and L5 (L3 reinforcement). Tasks with cooperative challenges	Tabata routines, active videos and curricular ABs	Homework assignments. Establishment of activities, passive learner.	Percents and probability	“Travelling the world”: homonyms, adverbs, v-words and b-words	The economy	Electricity and magnetism
13	All L (L4 reinforcement). Transfer activities and group reflection	Tabata routines, active videos and curricular ABs	Direct command. Imposing and setting tasks.				

Note: TPSR = teaching personal and social responsibility; AB = active breaks; L1 = respect for oneself and for the rights of others; L2 = participation and effort; L3 = personal autonomy; L4 = leadership and helping others; L5 = transfer out of school.

**Table 3 behavsci-12-00153-t003:** Example of TPSR and AB tasks carried out during programme implementation.

**TPSR**	**Example of task L1**	*“Team-building”*: to take active breaks in class as a group, and may not actively participate in the individual aspects, but always respecting fellow group members in carrying out the activities			
	**Example of task L2**	*“The Borg scale”*: a panel from 1 to 10 is placed during the active break, at which time the teacher indicates the level of intensity of the exercise, which should be performed by the students according to their self-regulation and personal effort			
	**Example of task L3**	*“Personal work plan”*: a short test of the contents covered is carried out, after which each student has to propose a task to improve the aspect where he/she got the worst mark and then present it in class			
	**Example of task L4**	*“Reciprocal teaching”*: the students are placed in one of the active breaks in pairs, one of them has a small script of the activity to be carried out and has to teach the other one how to do it, and then the roles are changed.			
	**Example of task L5**	*“Goose without words”*: a giant goose is made with squares (made by the pupils) and a goose is made with questions from a lower grade about their subject. In pairs, the older pupils help the younger pupils to solve the questions that are asked using mime.			
**AB**	**Tabata routine**	*(Mov.) 20”*: squats + jumping jacks + table push-ups (×2) *(R.) 10”*: rest	*(Mov.) 20”*: front stride + skipping + shoulder press (×2) *(R.) 10”*: rest		*(Mov.) 20”*: multisprings + abdominal on chair + up and down on chair (×2) *(R.) 10”*: rest
	**Active videos**	Trolls: Can’t Stop The Feeling (GoNoodle—Youtube)	Pump It Up—Fresh Start Fitness (GoNoodle—Youtube)		Smallfoot: Do The Yeti (GoNoodle—Youtube)
	**Curricular AB**	**Mathematics** *“Dividing with decimals”*: a division is written on the blackboard and the quotient of this division (only the decimal part) will indicate the number of seconds they will have to be conducting military marching in class	**Spanish Language** *“Words with b and v”*: the teacher pronounces words and the pupils have to indicate whether they are written with “b” or with “v”. If they get it right, they carry out 5 repetitions of the exercise of their choice and if they do not get it right, they carry out 5 multi-skips	**Social Sciences** *“The mountain systems”*: classify each system into its mountain group: inside the plateau (perform 8 squats), bordering the plateau (perform 8 table push-ups) and outside the plateau (perform 8 jumping jacks)	**Natural Sciences** *“Vertebrate and invertebrate animals”*: the teacher reads a paragraph about animals and each time an animal is mentioned, the pupils have to classify them into vertebrates (perform 6 climbers) or invertebrates (perform 6 front strides).

Note: TPSR = teaching personal and social responsibility; AB = active breaks; L1 = respect for oneself and for the rights of others; L2 = participation and effort; L3 = personal autonomy; L4 = leadership and helping others; L5 = transfer; Mov. = movement; R = rest.

## Data Availability

Data collected and analyzed during the study are available upon reasonable request.

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
