# Peer review of "Effects of a Hybrid Program of Active Breaks and Responsibility on the Behaviour of Primary Students: A Mixed Methods Study"

_behavsci, 2022, doi:10.3390/bs12050153_

Round 1
Reviewer 1 Report
First of all I would like to congratulate you for the research carried out.I think it is necessary and important today. On the other hand, I have seen in lines 89, 197 and 409 small errors related to citations,
they are quite minor but necessary changes.

Author Response
Thanks you to the reviewer, we have changed this formal aspects and improved the manuscript.
Reviewer 2 Report
The introduction provided the rationale for the study. It introduces activity breaks as well as teaching personal and social responsibility models in clear and concise way. The hybrid program seems promising way to improve student behaviour. Research design is based on a mixed methods approach, combining systematic observation methodology and semi-structured interviews, which seems suitable for researching such issue. It is based on one intervention carried out by one teacher compared to control group of one teacher. The instruments are described in adequate detail. The description of the procedure is however somewhat unclear. I think the format of the tables 2 and 3 makes them hard to read as well. Statistical analysis could be discussed with the use of instruments.
The main problem with the manuscript concerns the results section. The presentation of results based on OSTOR is very unclear. It is not clearly explained, how dendograms in Figure 1 were created. Figure 2 is even more baffling. It was very hard to figure out even the axis for the graph. The authors need to open up their results more as well as explain, how the figures should be interpreted.
The analysis of interview focuses on the number of mentions. I think a purely qualitative analysis of recognising issues and themes in teachers interview would make more sense. I think the analysis could be purely descriptive. It is unclear, were the 15 categories and four families formed inductively from the data. If they were, the result should focus on description of each category. The number of mentions in each category is not that interesting, as it is likely, that the number of mentions do not correlates with the perceived importance of each category.
The discussion seems reasonable and is based on the results as well as the previous studies. As most of the results seem to be in line with previous studies, highlighting how the results of this study can contribute something new to our understanding as well as practice might be needed.
Author Response
Thank you to the reviewer to give us the opportunity to improve this manuscript. We answer behind your appreciations:
The introduction provided the rationale for the study. It introduces activity breaks as well as teaching personal and social responsibility models in clear and concise way. The hybrid program seems promising way to improve student behaviour. Research design is based on a mixed methods approach, combining systematic observation methodology and semi-structured interviews, which seems suitable for researching such issue. It is based on one intervention carried out by one teacher compared to control group of one teacher. The instruments are described in adequate detail. The description of the procedure is however somewhat unclear. I think the format of the tables 2 and 3 makes them hard to read as well.
The procedure has been described in a more coherent and structured way to facilitate the reader's understanding. A sub-section called "2.4.1. Intervention" has been added to identify what was done in the GC and EG during the development of the programme. In addition, a sentence has been written before tables 2 and 3 in order to clarify the content of both tables.
Statistical analysis could be discussed with the use of instruments.
In the statistical analysis section it has been added some more sentences explaining the use of observational instruments.
The main problem with the manuscript concerns the results section. The presentation of results based on OSTOR is very unclear. It is not clearly explained, how dendograms in Figure 1 were created.
The figure1 has been explained in detail starting in 2.5 section in order to make it clear to the reader. Later, in the Results section a new paragraph has been written down explaining the dendogram figure 1.
Figure 2 is even more baffling. It was very hard to figure out even the axis for the graph. The authors need to open up their results more as well as explain, how the figures should be interpreted.
The figure 2 has been also explained in more detail as:
“Figure 2 shows the Event Time Plot of all the events registered throughout the intervention. The x axis is the temporal line of frames separated by periods of 5 minutes (almost 8.000 video frames) by the solid vertical blue lines. The y axis refers to the codes of behaviours observed (e.g. nº 41 concerns to SUG, AUT (Suggested and Autonomy)). The vertical solid line in the center separates EG and CG. We highlighted rectangles the codes of behaviors that appear, whether more density or a lack, in order to offer a more detailed analysis of the performance of the two teachers.
The analysis of interview focuses on the number of mentions. I think a purely qualitative analysis of recognising issues and themes in teachers interview would make more sense. I think the analysis could be purely descriptive. It is unclear, were the 15 categories and four families formed inductively from the data. If they were, the result should focus on description of each category. The number of mentions in each category is not that interesting, as it is likely, that the number of mentions do not correlates with the perceived importance of each category.
Thank you for your apreciations, we have changed this sectiond and taking into account your recommendations to do more qualitative analysis and not including the categories or codes like quantitative aspects.
The discussion seems reasonable and is based on the results as well as the previous studies. As most of the results seem to be in line with previous studies, highlighting how the results of this study can contribute something new to our understanding as well as practice might be needed.
Thank you for your considerations. We have also included at the end of the discussion the relevance to use observational methodology with the technique to detect of T-pattern analysis (TPA) that allows to refine so much results as we have obtained in these studies.
Reviewer 3 Report
- This manuscript has been written in an academically sound style. The topic and issues discussed are also important and practical. However, some modifications are still needed to make the manuscript suitable for publication.
- As this is a quasi-experimental study, the experimental and control groups were intentionally instead of randomly selected. Yet I still suggest that the authors could explain how to assume at least these two groups had similar background settings in terms of the significant variables related to the teachers' teaching style, learning habits of the students, and others.
- There were many short paragraphs in the manuscript, e.g., lines 60 to 82. I suggest the authors re-organize them, presenting the context more comprehensively.
- Both AB and TPSR and their hybrid were examined in previous studies. I suggest the author explicitly clarifies the existing research gap and the uniqueness of this study.
- As mentioned in the manuscript, adolescents are considered those with externalizing and internalizing problems. Were the students in both groups “under control” or following the courses' design? Did the teachers do anything to confirm those programs operated as anticipated?
- For those examples listed in Table 3, what are the respective objectives in the TPSR?
- Figures 2, 3, 5 and 6 were too small to read. Please enlarge the images.
Author Response
Thank you to the reviewer for your appreciations. We ask behind your comments:
1. This manuscript has been written in an academically sound style. The topic and issues discussed are also important and practical. However, some modifications are still needed to make the manuscript suitable for publication.
We really appreciate all the comments the reviewer has added to make more comprehensive this manuscript and expect the changes are suitable with your suggestions.
2. As this is a quasi-experimental study, the experimental and control groups were intentionally instead of randomly selected. Yet I still suggest that the authors could explain how to assume at least these two groups had similar background settings in terms of the significant variables related to the teachers' teaching style, learning habits of the students, and others.
We asume that both groups start form similar behavior patterns because both groups of students belong to the same educational level and school, they have similar socioeconomic characteristics and the exclusion criteria used were that one of them had had previous experience with this kind of methodology. In addition, the teachers used a methodology based on conventional teaching and had a very similar previous experience of 10 and 12 years. Furthermore, the data recorded in the first lessons allow us to appreciate that the differences in behavior were minimal between the control and experimental groups or non-existent (lessons 2 and 3) after the initial training received to the experimental group teachers. In the first lesson some patterns are shown thanks to the indications received innitially and that do not occur again after several lessons with more continuous information from the training team.
3. There were many short paragraphs in the manuscript, e.g., lines 60 to 82. I suggest the authors re-organize them, presenting the context more comprehensively.
The paragraphs with three or less lines have been re-organized, presenting the context more comprehensively.
4. Both AB and TPSR and their hybrid were examined in previous studies. I suggest the author explicitly clarifies the existing research gap and the uniqueness of this study.
However, there is not still any study which had hybrided both methodologies looking for its viability and the impact in students values and PA.
5. As mentioned in the manuscript, adolescents are considered those with externalizing and internalizing problems. Were the students in both groups “under control” or following the courses' design? Did the teachers do anything to confirm those programs operated as anticipated?
Yes, they did. Student behaviours were observed next to the teacher patterns, as it is showed in figures 1, 2, 3, and 4, highlighting an unbalance and a misaligned and reproductive response from the students in control group (lines 309-311). Students from experimental group showed an autonomous response (lines 301-302). Regarding teachers in control group, he imposed task instruction and set them by himself, without giving alternatives and giving negative evaluations (306-309). Teacher from experimental group suggested the intervention, giving autonomy and positive evaluation (line 299-300). Lately, some of these ideas are mentioned in the discussion section and finally, at the end of the manuscript with the conclusions in the last three paragraphs.
6. For those examples listed in Table 3, what are the respective objectives in the TPSR?
It has been added what do L1-L5 mean in notes for Table 2 and 3. They are the TPSR levels which are the respective objectives. Thank you for make us aware of this detail.
7. Figures 2, 3, 5 and 6 were too small to read. Please enlarge the images.
The six Figures have been enlarged in order to make them easier to read.
Reviewer 4 Report
The topic can be of interest, but as it just include 3 teachers and their students on cant make conclusions as you are doing.
Too small sturdy ot make arguments as you are doing
NO self criticism either
Ref from 2021 can be improved
Author Response
Thank you to the reviewer for your comments.
1. The topic can be of interest, but as it just include 3 teachers and their students on cant make conclusions as you are doing. Too small sturdy ot make arguments as you are doing
We respect the opinion of the reviewer, however, qualitative studies with interviews are used to having a short number of participants. For example: Manzano-Sánchez, D.; Gómez-Mármol, A.; Valero-Valenzuela, A. Student and Teacher Perceptions of Teaching Personal and Social Responsibility Implementation, Academic Performance and Gender Differences in Secondary Education. Sustainability 2020, 12, 4590. https://doi.org/10.3390/su12114590 with two teachers for the interviews.
2. NO self criticism either
Modifications have been made in the last three paragraphs of the discussion to give greater coherence to the arguments of the discussion and the results found, highlighting the limitations of the study and the caution in considering and interpreting both the results and the conclusions of the present research (Lines 445-479).
3. Ref from 2021 can be improved
The 2021 references have been improved, including others such as:
Pastor-Vicedo, J. C.; Prieto-Ayuso, A.; López Pérez, S. ; Martínez-Martínez, J. Active Breaks and Cognitive Performance in Pupils: A Systematic Review. Apunt. Educ. Fis. 2021, 146, 11-23. https://doi.org/10.5672/apunts.2014-0983.es.(2021/4).146.02
Shen, Y.; Martinek, T.; Dyson, B. P. Navigating the processes and products of the teaching personal and social responsibility model: A systematic literature review. Quest. 2022, 74(1), 91-107.
Simonton, K. L.; Shiver V. N. Examination of elementary students’ emotions and personal and social responsibility in physical education. Eur. Phy. Educ. Rev. 2021, 27(4), 871-888. https://doi.org/10.1177/1356336X211001398
Zerf, M.; Kherfane, M. H.; Bouabdellah, S. B. A. Classroom routine frequency and their timing practice as critical factor to build the recommended primary school active break program. Retos. 2021, 41, 434-439. http://dx.doi.org/10.47197/retos.v0i41.77808
Reviewer 5 Report
The object of the study is particularly interesting in the field of educational research and, in particular, in the model of personal and social responsibility of teachers applied to physical education.
The research develops a quasi-experimental design, with the aim of verifying the effects of the application of the Active Break program, oriented to the promotion of values education.
The techniques applied were systematic observation (group -students- and individual -teachers-) to collect data on teaching and communication strategies during the implementation of the program, and interviews to know their perceptions on the results obtained in the teaching sessions. Both techniques are applied with methodological rigor.
It is suggested, as a minor recommendation, to specify in detail the objective of the study and justify its relevance in the specific field of behavioral sciences.
Author Response
Thank you for your comments.
We have explain the importance to use active methodologies (like in this study) to improve behavioral in students.
Round 2
Reviewer 4 Report
Although the authors have made several improvements
The article is of minor interest and also with just 2 teachers involved it is a very small scale research
More like a case study
The structure and arguments must be clearer and also the self crisisims and own limitations can be explored
Fig 3+4 is impossible to understand